# Redox and Anti-Inflammatory Properties from Hop Components in Beer-Related to Neuroprotection

**DOI:** 10.3390/nu13062000

**Published:** 2021-06-10

**Authors:** Gustavo Ignacio Vazquez-Cervantes, Daniela Ramírez Ortega, Tonali Blanco Ayala, Verónica Pérez de la Cruz, Dinora Fabiola González Esquivel, Aleli Salazar, Benjamín Pineda

**Affiliations:** 1Laboratory of Neurobiochemistry and Behavior, National Institute of Neurology and Neurosurgery, 14269 Mexico City, Mexico; guigvace@gmail.com (G.I.V.-C.); drmz_ortega@hotmail.com (D.R.O.); tonaliblaya@gmail.com (T.B.A.); veped@yahoo.com.mx (V.P.d.l.C.); dinora.gonzlez@gmail.com (D.F.G.E.); 2Laboratory of Neuroimmunology, National Institute of Neurology and Neurosurgery, 14269 Mexico City, Mexico; ajsalazar27@gmail.com

**Keywords:** hop, antioxidant, xanthohumol, prenylflavonoids, beer

## Abstract

Beer is a fermented beverage widely consumed worldwide with high nutritional and biological value due to its bioactive components. It has been described that both alcoholic and non-alcoholic beer have several nutrients derived from their ingredients including vitamins, minerals, proteins, carbohydrates, and antioxidants that make beer a potential functional supplement. Some of these compounds possess redox, anti-inflammatory and anticarcinogenic properties making the benefits of moderate beer consumption an attractive way to improve human health. Specifically, the hop cones used for beer brewing provide essential oils, bitter acids and flavonoids that are potent antioxidants and immune response modulators. This review focuses on the redox and anti-inflammatory properties of hop derivatives and summarizes the current knowledge of their neuroprotective effects.

## 1. Generalities

Beer is an alcoholic natural beverage, not distilled, from barley extract. The traditional ingredients for brewing beer are barley (malt), hops (*Humulus lupulus* L.), yeast, and water. The brewing steps and brewing conditions are essential to generate specific beer characteristics and quality. In general, the first step of brewing is the malting process which consists of germinating the barley grains in water; subsequently, the flavor and color of the beer are given by roasting the grains. Afterward, a sweet mixture is obtained by grinding the cereal at the appropriate temperature. After boiling, the female inflorescences of the hop plant (*Humulus lupulus* L., Cannabaceae) are added, giving a colloidal strength to beer. It should be noted that beer bitterness is attributable to the hop plant. The final step of brewing is fermentation, which begins just after yeast addition. The most used yeast strains are *Saccharomyces cerevisiae* (stout, ale and porter beers) and *Saccharomyces uvarum* (lager beers); however, non-conventional yeast strains have also been used [1,2]. According to the fermentation parameters used during brewing, beers are classified into two major classes: top and bottom-fermented beers. In top-fermented beers (ale-type beers), yeast growth occurs in the upper part of the container and the fermentation temperature ranges between 16 °C and 24 °C; while in the bottom-fermented beers (lagers), yeast growth occurs in the base of the container and the fermentation temperature ranges between 8 °C and 15 °C [3]. 

The moderate consumption of beer has been related to beneficial human health effects; however, harmful effects have been described by the consumption of beverages with high alcohol content. Low-moderate intake of alcohol is considered up to one drink per day for women and up to two in men (considering that typically a can of beer has 330 mL containing about 4% *w*/*v* alcohol). Drinking beer contributes to the intake of carbohydrates and it can be an important risk factor for obesity and overweight; nevertheless, an inconsistency between beer consumption and obesity has been observed due to the contribution of other risk factors such as individual diet, physical activity, or the consumption of other alcoholic beverages [2,4,5]. On the other hand, it has been described that the regular and moderate consumption of beer confers cardiovascular protection similar to that of wine; and also reduces mortality both in healthy adults and in cardiovascular patients [6]. However, alcohol abuse is harmful to several human organs and a major social and health problem associated with addictions, accidents, violence, and crime [7,8,9,10]. Additionally, moderate beer consumption has been associated with lipid profile improvement in blood plasma; reduction in leukocyte adhesion molecules and inflammatory markers improving the prognosis of cardiovascular diseases such as atherosclerosis and thrombotic ailments. Other studies regarding beer consumption showed better beneficial effects in vascular endothelial function and pressure wave reflections compared to other kinds of alcoholic beverages [2,6,11,12]. Clinical studies have suggested that the moderate consumption of beer is beneficial for human health, mainly due to the phenolic compounds with antioxidant and anti-inflammatory properties.

Beer is a beverage rich in phenolic compounds derived from hop (30%) and malt (70–80%) [13]. It has been described that its phenolic acid content ranges from 3 to 12 mg/L and its total polyphenol content ranges from 74 to 256 mg/L in 34 different lager beers brewed in various geographic locations [14]. Specifically, the resin secreted by lupulin glands contains bitter acids, essential oils, and prenylated flavonoids (Figure 1) that have received particular attention because of their high antioxidant, anti-inflammatory and anticarcinogenic effects, among others. Recently, a study focused on the biosynthesis and regulation of these compounds demonstrated an increased expression of transcription factors and structural genes in lupulin glands after leaf development, which correlated with increased levels of bitter acids and prenylflavonoids in these glands [15]. Derived compounds from hop are used as a source of bitterness, herbal aroma and natural preservative. 

Essential oils represent 0.5–3.0% from hop dry weight and can be classified into hydrocarbons (50–80% of total oil) such as limonene, myrcene, and pinene; oxygenated compounds (30% of total oil), such as linalool, caryophyllene, geraniol, and farnesol; and sulfur-containing compounds (around 1% of total oil) [16]. The bitter acids found in hops are α-acids (humulone, cohumulone, and adhumulone) and β-acids (lupulone, colupulone, and adlupulone) that differ from each other because the β-acids contain an extra isoprenyl group [17]. After thermal isomerization during the brewing process, α-acids are transformed into iso-α-acids (which occurs favorably at temperatures around 100–130 °C and pH of 8–9) which are considered the primary drivers of hop-derived beer bitterness, meanwhile, β-acids are oxidized during the brewing process [16,18,19]. Additionally, hops are a source of polyphenols; however, brewing malt is the major source of these compounds [20]. The main hop polyphenols are flavonoids, catechins, phenolic acids, prenylated-chalcones and proanthocyanidins. Prenylated flavonoids combine a flavonoid skeleton with a lipophilic prenyl side-chain, increasing the lipophilicity of flavonoids [21]. Xanthohumol is the principal prenylated chalcone in the lupulin glands (0.1–1% in dry weight), but it is a minor prenylflavonoid in beer since it contains a free 2′-hydroxy group which, after thermal isomerization during the brewing process, produces its corresponding flavanone isoxanthohumol (ranging from 0.04 to 3.44 mg/L), which is the most abundant flavonoid in beer [22,23,24]. A pharmacokinetic study performed in healthy subjects (24 males of 31 ± 2 years old and 24 females 35 ± 2 years old) showed that after a single oral dose of 20, 60 or 180 mg of xanthohumol, the maximum xanthohumol concentrations were 33, 48 and 120 mg/L, respectively, with isoxanthohumol being the main circulating metabolite [25]. Beer is the principal dietary source of xanthohumol and related prenylflavonoids [23], their health benefits as well as the mechanism by which these hop derivatives act are the subject of several investigations. 

The next section describes the current knowledge on redox, anti-inflammatory and immunomodulatory effects of hop components in experimental models, mainly focused in xanthohumol and its derivative isoxanthohumol, which represents the major prenylflavonoid in beer. Subsequently, a comprehensive compilation of human beer consumption effects as well as hop components on redox, anti-inflammatory and immunomodulatory markers are summarized. Some novel information of selected beer hop components on neuroprotection is also provided.

## 2. Redox Properties from Hop Derivatives on Beer

Free radicals and reactive oxygen species (ROS) are highly reactive compounds that easily oxidize other molecules by taking up electrons [26]. ROS are normally produced during cell metabolism mainly in the mitochondrial electron transport chain, which has been described as an important role of ROS in the regulation of metabolism, cell cycle and survival, among others [26,27]. However, when excessive ROS are produced, the oxidation of biomolecules such as lipids, proteins, and nucleic acids occurs, leading to cellular stress and oxidative damage. The role of oxidative stress in the progression of several diseases and aging has been widely reported, which is the reason why much investigation is focused on the search for compounds that attenuate the oxidation processes [26,27,28,29].

The benefits of hop derivatives contained in beer are associated with their ability to modulate the redox environment since they can scavenge a wide range of reactive oxygen species, but it has also been described that hop components can modulate the expression and activity of antioxidant enzymes and the glutathione (GSH) levels, thus protecting against toxic stimulus.

Several studies have evaluated the antioxidant capacity of beer varieties, finding that their redox activity depends on their polyphenolic amount. The polyphenolic content varies depending on the type of beer from 366 gallic acid equivalents (GAEs) mg/L in alcohol-free beers to 875 GAE mg/L in bock beers [30]. In general, beers are able to scavenge free radicals in ferric antioxidant power (FRAP) and 2,2-azino-bis (3-ethylbenzothiazoline-6-6sulphonic acid) (ABTS) assays, among others, as well as to prevent DNA oxidation [30,31,32,33,34]. In recent years, many research groups have studied the redox properties of beers characterizing the fractions obtained during brewing or testing individual components.

In 2009, Gerhauserin [35] analyzed the scavenging capacity of 51 phenolic beer compounds. The redox activity of polyphenols is determined in part by the OH groups found in the aromatic nucleus, mainly in the C4′ and C3′ positions. General radical scavenging potential was determined by the reaction with 1,1-diphenyl-2-picrylhydrazyl (DPPH), the group of proanthocyanidins and flavonols that had a better scavenging potential considering their half maximal scavenging concentration (SC_50_ 7.6–16 µM and 10.4–23.7 µM, respectively). When the phenolic compounds were exposed to a system able to chemically generate superoxide, the best scavenging profile was shown by protoanthocyanidins, catechins, flavonols and flavones. However, when the superoxide was produced by an enzymatic reaction, the catechins and proanthocyanidins were identified as the most potent scavengers of the beer phenolic compounds tested. Furthermore, xanthohumol, saponaretin and 4-ketopinoresinol were the only compounds that showed the inhibition of superoxide production, stimulated with phorbolmyristate acetate (PMA) in HL60 human promyelocytic leukemia cells with an IC_50_ of 5.5 µM, 4.7 µM and 7.1 µM, respectively. Proanthocyanidins, catechins, flavonols, xanthohumol, and isoxanthohumol showed a better peroxyl radical scavenging potential than Trolox (vitamin E analogous with high antioxidant capacity) at 1 µM. The hydroxyl radical scavenging profile was also evaluated in phenolic beer compounds, most of them were able to scavenge this radical. However, xanthohumol was the best compound to scavenge hydroxyl radical followed by caffeic acid, myricetin and cinnamic acid when compared to Trolox [35].

Hop bitter acids, humulone and lupulone have a radical scavenging activity with an IC_50_ around 2–3 × 10^−5^ M and lipid peroxidation inhibitory activity (7.9 × 10^−6^ and 3.9 × 10^−5^, respectively), these effects were similar to those obtained with α-tocopherol and ascorbic acid. It has been suggested that the 5-hydroxyl group of bitter acids is the active site for radical scavenging activity since those analogs lacking this group do not have this activity [36].

One of the most studied and the most abundant prenylated chalcones (open C-ring flavonoids) from hops is the xanthohumol [22]. Xanthohumol is 9-fold and 3-fold more effective than Trolox in scavenging peroxyl and hydroxyl radicals, respectively. However, isoxanthohumol was more potent than xanthohumol for peroxyl radical scavenging and showed the same effect as Trolox for scavenging hydroxyl radicals. According to these data, xanthohumol (1.5 and 3 µM) reduced the electron spin resonance signal intensity of hydroxyl radical formation in a cell-free system (H_2_O_2_/NaOH/DMSO) [37]. Xanthohumol inhibited the production of superoxide in the xanthine/xanthine oxidase system with an IC_50_ 27.7 ± 4.9 µM, while xanthohumol showed an IC_50_ of 2.6 ± 0.4 µM for superoxide scavenging in TPA-stimulated HL-60 cells [38]. Both xanthohumol and isoxanthohumol reduced nitric oxide (NO) production (IC_50_ 12.9 ± 1.2 µM and 21.9 ± 2.6 µM, respectively) induced by lipopolysaccharide (LPS) in RAW 264.7 mouse macrophages [38]. An additional study showed that xanthohumol inhibited the production of NO with an IC_50_ 8.3 µM in RAW 264.7 macrophages exposed to LPS and IFN-γ [39]. According to these data, it was reported that the ethyl acetate soluble fraction of *Humulus lupulus* L. that contains chalcones including xanthohumol inhibits the production of NO by suppressing the expression of iNOS in RAW 264.7 cells exposed to LPS and IFN-γ [40]. 

Additionally, xanthohumol was able to neutralize around 60% of ABTS free radicals and decrease 30% of the thiobarbituric acid-reactive substances (TBARS) formation induced by free radicals in erythrocyte ghosts [41]. Furthermore, xanthohumol and other prenylated chalcones found in beers were able to inhibit the oxidation of human low-density lipoprotein (LDL), the formation of TBARS, and the oxidation of tryptophan residues in LDL induced by Cu^2+^ [42]. An additional study found that xanthohumol decreased lipid peroxidation in rat liver microsomes induced by Fe^2+^-ascorbate or *tert*-butyl hydroperoxide (TBH). The protective effect of xanthohumol (0.1 and 0.5 µM) was observed in PC12 cells from H_2_O_2_-induced and 6-hydroxydopamine-induced cellular damage due to a reduction in both ROS production and caspase-3 activity [41]. It was also shown that 5 µM and 10 µM of this chalcone decreased lactate dehydrogenase activity (by around 25 and 50%, respectively) in the primary cultures of rat hepatocytes treated with TBH; this anti-cytotoxic effect showed by xanthohumol was related to its antioxidant activity [43]. 

The protective effects shown by xanthohumol can be in part related to the activation of nuclear factor erythroid-2-related factor 2 (Nrf2) and the expression of the Nrf2-driven antioxidant/detoxifying genes. Under normal conditions, Nrf2 is bonded to the Kelch-like ECH-associated protein 1 (Keap1) in the cytoplasm, and Nrf2 is ubiquitinated for its proteasomal degradation; however, under oxidative stress conditions, the Nrf2-Keap1 complex is dissociated and Nrf2 is translocated to the nucleus where it binds to the antioxidant-responsive element (ARE), thus promoting the transcription of antioxidant-protective genes such as heme oxygenase-1 (HO-1), NAD(P)H: quinone oxidoreductase 1 (NQO1), thioredoxin (Trx1), thioredoxin reductase 1 (TrxR1), glutamate cysteine ligase (GCL). It has been shown that xanthohumol increases the content of nuclear Nrf2 and decreases its cytosolic localization, indicating its nuclear translocation and promoting the antioxidant response. Moreover, it was demonstrated that xanthohumol (0.5 µM) increased the mRNA expression of HO-1, NQO1, Trx1, TrxR1, GCL after 6 and 12 h, with HO-1 and TrxR1 mRNAs being the most highly transcribed (by around 2.8- and 2.3-fold). In line with the upregulation of phase II enzymatic expression, the protein and activity of these enzymes as well as GSH levels were also increased by xanthohumol [41]. Likewise, it has been suggested that the anti-inflammatory effect shown by xanthohumol could be related to its Nrf2-ARE signaling modulation [44].

The antioxidant cellular modulation by hop components contributes to the beneficial effects of beer intake. There is in vivo evidence that beer intake (0.5 mL/day) reduces lipid peroxidation and TNFα over-expression induced by the oral administration of aluminum in mice. These effects were related to the restoration of the antioxidant enzymatic expression of catalase and superoxide dismutase [45]. Another study demonstrated that treatment with beer fractions derived from the brewing process reduced oxidative stress as well as the senescence induced by H_2_O_2_ in dental-derived stem cells and human intestinal epithelial lines (Caco-2 cells) [13]. Moreover, the consumption of beer with or without alcohol restored the enzymatic activity of complex I and IV and prevented the oxidation of coenzymes Q_9_ and Q_10_ in the heart and liver mitochondria from rats treated with adriamycin [46]. In addition to redox modulation, the immunomodulatory and anti-inflammatory effects of the hop components were extensively studied and are described in the following section.

## 3. Anti-Inflammatory and Immunomodulatory Effects of Beer Compounds

Crosstalk between the oxidative and pro-inflammatory signals has been described, making these two important factors in the progression of age-related disorders. Inflammation is the response elicited by the immune system to counteract noxious agents, which is a complex process that requires the precise balance of intracellular and extracellular signals, first elicited by the innate immune system cell populations including neutrophils, macrophages, dendritic cells and natural killer cells, later reinforced by the components of the adaptive immune response, T and B lymphocytes [47]. The synchronic orchestration of both innate and adaptive immune responses leads to the beginning and control of inflammation, and the reestablishment of the organism homeostasis. However, dysregulated inflammation is associated with the development of age-related disorders such as cancer, neurodegenerative diseases, and chronic inflammation disorders [48]. 

Thus, in addition to the search for natural compounds with antioxidant properties, finding anti-inflammatory or immunomodulatory characteristics in the same molecules would be ideal. In light of this, both the anti-inflammatory role of isolated beer compounds as well as the effect of moderate beer consumption on the immune system have been investigated. It was reported that beer consumption in healthy rats, considering a daily ethanol consumption of 1.16 g/kg, has no impact on the relative abundance of the different lymphocytic populations [49]. Additionally, the positive effects of beer consumption have been demonstrated, since this prevented the formation of atherosclerotic plaques in addition to the decrease in the expression of the endothelial adhesion molecules: intercellular adhesion molecules (ICAMs) and vascular cell adhesion molecules (VCAMs), together with a decreased expression of the nuclear factor κB (NF-κB, an important transcription factor that drives the production of pro-inflammatory signals), indicating an anti-inflammatory effect in atherosclerotic rats [50]. Moreover, beer consumption promoted anti-tumor immune responses and increased the number of T lymphocytes on tumor-bearing rats [51]. 

As mentioned above, the female cones of hop are a major source of beer polyphenols and bioactive compounds. Thus, the effect as immunomodulators of the whole hop extracts in the immune system has been studied (Figure 2). In vitro treatments using hop extracts on LPS-stimulated macrophages and peripheral blood mononuclear cells (PBMCs) decreased the levels of the proinflammatory cytokines interleukin (IL)-1β, IL-6, tumor necrosis factor (TNF)-α and the monocyte chemo-attractant protein (MCP)-1; in addition, they decreased the production of nitric oxide (NO), a diffusible ROS involved in vasodilation and immune infiltration [52,53,54]. The anti-inflammatory activity of hop extracts is attributed to the inhibition of the NF-κB activation. Hop extracts also decreased the activity of the enzyme cyclo-oxygenase 2 (COX-2) leading to the reduction in the inflammatory mediator prostaglandin E2 (PGE2) in a mouse model of zymosan-induced arthritis [55]. Likewise, these anti-inflammatory effects of hop extracts (1–10 μg/mL) have been described in epithelial cells stimulated with viral double-stranded RNA where they decreased the production of TNF-α and the thymic stromal lymphopoietin (TSLP), a cytokine involved in allergic reactions, thus suggesting a role of hop extracts on the modulation of allergic responses [56]. More refined hop extracts contain only the α-acids, β-acids and iso-acids, also known as bitter acids. The bitter acids (1–50 mg/kg) prevented brain inflammation and depressive behavior in mice exposed to LPS-induced brain intoxication [57]. Moreover, these molecules also decreased the production of TNF-α and IL-1β in TNF-α-stimulated fibroblasts and activated the hepatic stellate cells, suggesting a role of bitter acids in the prevention of fibrosis [58,59].

Iso-α-acids have been widely studied in the modulation of the innate immune system within the brain and its implications in neurodegenerative disorders. Iso-α-acids promoted the phagocytic activity of microglial cells in vitro together with a shift from a pro-inflammatory towards an anti-inflammatory phenotype, as evidenced by reduced levels of NO, TNF-α, IL-1β, IL-6, IL-12 and MCP-1, while the number of the anti-inflammatory microglial marker CD206 was increased [60,61,62,63]. In addition to its role in preventing neuroinflammation, iso-α-acids prevented the hepatic damage observed in non-alcoholic steatohepatitis induced by a Western diet, reducing hepatic fibrosis, inflammation and oxidative damage [64]. In the case of β-acids, it has been observed to have an inhibitory effect on the activation of NF-κB, thus inhibiting tumor generation in mice exposed to 12-*O*-tetrahydrophorbol 13-acetate [65]; however, the role of separated β-acids in immune activation remains poorly explored. 

To specifically identify which beer compounds are responsible for these effects, several studies have focused on studying them individually (Table 1).

### Immunomodulatory Effect of Xanthohumol

The isolated compounds of hop extracts have been tested in several models. Regarding the immune system and inflammatory process, xanthohumol and its derivative isoxanthohumol have been described as potent anti-inflammatory molecules. In vitro studies demonstrated that both xanthohumol and isoxanthoumol, at concentrations ranging between 0.5 and 20 µM, suppressed the production of pro-inflammatory cytokines such as IL-1β, IL-6, IL-12, MCP-1, and TNF-α, together with a decreased expression of the inducible nitric oxide synthase (iNOS) and reduced levels of NO on macrophages and monocytes stimulated with LPS or interferon-γ (IFN-γ) [44,66,67,68,69]. The mechanisms through which xanthohumol exerts its anti-inflammatory effects have been attributed to its intervention at different levels of the macrophage inflammatory signals. Xanthohumol interferes with the Toll-like receptor 4 (TLR-4) signaling by binding directly to its co-receptor, the myeloid differentiation protein 2 (MD-2) and decreases TLR-4 expression [68,70]. Additionally, it blocks the interferon response factor 1 (IRF1)/STAT1 cascade [66] and inhibits the activation of the NF-κB [44,66,67,69], which also leads to reduced levels of the inflammasome components NLRP-3 and caspase-1, and finally to the reduction in IL-1β levels [67]. Conversely, the mechanism of action of isoxanthohumol in the pro-inflammatory signaling cascade only points to the inhibition of the NF-κB cascade but no changes in TLR-4 expression have been described [68,75]. 

The anti-inflammatory role of xanthohumol has been observed in vitro in different tissue resident cell types that contribute to the innate immune response such as chondrocytes, intestinal epithelial cells, hepatocytes and hepatic stellate cells. In these cells, xanthohumol decreased the production of inflammatory markers NO, IL-6, IL-8, TNF-α, MCP-1, COX-2, and PGE2 [74,75,76]. In contrast, a deleterious role of xanthohumol has been reported in dendritic cells, another cellular subset of the innate immune response. In this context, xanthohumol, at 50 µM, induced apoptosis triggered by caspase-8 in a mechanism dependent on the activity of the enzyme acid sphingomyelinase that led to an increased ceramide production in vitro [71].

The anti-inflammatory role of xanthohumol in vivo has been tested in models of acute and chronic inflammation in different organs. The topical administration of xanthohumol prevented skin inflammation in oxazolone-induced dermatitis [69] and its intraperitoneal administration showed anti-inflammatory and antioxidant effects as well as the prevention of fibrosis on acute lung injury induced by LPS injection [67]. 

Supplementation of a standard rodent diet with xanthohumol in different murine experimental models including ischemia/reperfusion, carbon tetrachloride acute liver toxicity and non-alcoholic steatohepatitis was shown to prevent the increase in proinflammatory markers IL-1α, IL-6, MCP-1, TNF-α and ICAM-1; it also decreased the levels of the transforming growth factor β (TGF-β), TIMP1, collagen-1 and α-smooth muscle actin (α-SMA) and contributed to the prevention of liver fibrosis in these models [76,78,79]. The oral administration of xanthohumol decreased the inflammatory markers and the severity of dextran sulfate sodium-induced colitis where TNF-α, IL-1β and COX-2 decline was associated with xanthohumol-induced inhibition of NF-κB signaling [75]. Moreover, xanthohumol-supplemented beverages given to animals decreased the levels of the pro-inflammatory markers IL-1β and NO in addition to the decrease in the vascular endothelial growth factor (VEGF) and the number of new blood vessels in a skin wound healing model, suggesting that xanthohumol also modulates the angiogenic process [80]. These results are in line with previous studies showing that both xanthohumol and isoxanthohumol prevented the assembly of capillary-like structures from endothelial cells by inducing apoptosis on these cells [77]. The anti-inflammatory role of isoxanthohumol has been demonstrated in models of high-fat diet-induced insulin resistance where it decreased levels of the pro-inflammatory cytokines IL-1β and TNF-α and improved the glucose tolerance; these effects were attributed to isoxanthohumol-induced changes in the abundance and diversity of mice microbiota [81]. 

Despite the described mechanisms of xanthohumol in the function of the innate immune system cells, the role of this chalcone on the modulation of adaptive immune response remains poorly understood. In vitro studies evidenced that xanthohumol exerts an antiproliferative role on concanavalin-A or IL-12-activated T lymphocytes, an effect which was accompanied by cell cycle arrest but also with the increase in apoptotic cells [72,73] and it has been attributed to the inhibition of STAT and NF-κB signaling as occurred in cells of the innate immune response [72,73]. Moreover, xanthohumol suppressed the cytotoxic activity and the production of IL-2, TNF-α and IFN-γ on T lymphocytes suggesting the inhibition of pro-inflammatory T cell responses (TH1) in vitro [72]. However, xanthohumol showed opposite effects on the stimulation of TH1 lymphocytes in vivo on a breast cancer mouse model, where xanthohumol induced the reduction in tumor mass, the increase in cytotoxic lymphocytes together with an increase in IFN-γ and IL-2. Additionally, it produced a reduction in the lymphocyte-produced anti-inflammatory cytokines IL-4 and IL-10, favoring an anti-tumor TH1 phenotype instead of an anti-inflammatory TH2 phenotype on T lymphocytes [82]. This contradictory evidence suggests that the role of xanthohumol is beyond its anti-inflammatory properties, as further research using different models of immune activation would reveal a more complex role of xanthohumol in the modulation of the immune response.

The role of 8-prenylnaringenin on the inflammatory process has also been studied, demonstrating that this compound can reduce the expression of pro-inflammatory markers such as IL-12, TNF-α and NO together with a decrease in the production of ROS and PGE2 [69,83]. Furthermore, contrary to xanthohumol and isoxanthohumol, 8-prenylnaringenin modulates the angiogenic processes by inhibiting the cell death of vascular endothelial cells and promoting the formation of capillary structures in wound healing models [77].

## 4. Effect of Human Beer Consumption on Redox Environment and Immunomodulation

The role of moderate beer consumption has been studied in the immunological and redox markers of healthy subjects and people with cardiovascular risk. The favorable beer consumption effects can be related to the mechanisms previously described in experimental models. In a randomized feeding trial comparing moderate beer intake (660 mL/day, containing 30 g of ethanol and 1209 mg of polyphenols) with non-alcoholic beer (900 mL/day containing 1243 mg of total polyphenols), reduced serum concentration of inflammatory biomarkers and also beneficial effects on the cardiovascular system with better results than distilled beverages such as gin (100 mL/day, containing 30 g of ethanol) were observed, probably because of the polyphenolic content of beer [93]. Another study showed that non-alcoholic beer intake for 45 days followed by hop supplementation (400 mg/days) decreases lipid peroxidation and carbonyl groups while it increases the GSH and α-tocopherol content in the blood. Interestingly, the inflammatory parameters IL-6 and complement C3 fraction decreased just after hop supplementation [94].

Comparisons between social and problem alcohol drinkers demonstrated that problem drinkers present lower levels of both serum IL-6 and IL-1 receptor agonist, (pro-inflammatory and anti-inflammatory markers, respectively) suggesting that these subjects have a blunted immune response compared to social drinkers, these differences were related to anxiety and motivation behavior in the studied subjects [95]. In the short term, 30 min after consumption, beer decreased the cytotoxic activity of peripheral lymphocytes stimulated with IL-2 or phytohemagglutinin of healthy subjects [86]; this would seem like a detrimental role of beer consumption on the immune response immediately after ingestion, nevertheless, blood lymphocytes were more resistant to radiation-induced damage from 30 min to 4.5 h after beer ingestion [87]; this could be beneficial in the case of cancer patients receiving radiotherapy. Moderate (330 mL for women, 660 mL for men) beer consumption in healthy subjects did not affect the serum levels of the adhesion molecules ICAM-1 and E-selectin [88] nor the levels of cytokines or complement proteins involved in the serum immune response [90]. Moreover, as part of a study to determine the effect of moderate beer consumption on immunocompetence in healthy adults, sixty subjects (29 women and 31 men) between 25 and 50 years consumed, after a month of abstinence, one or two cans of beer as part of a regular diet during a month. An increased number of peripheral leukocytes, neutrophils, basophils, and CD3+ lymphocytes was observed in women while in men it was only a trend [8,89]. There were no differences in CD4+, CD8+, CD19+, TNF-α, and IL-6. Nevertheless, an increase in IgG, IgA, and IgM values was observed both in men and women [8]; ex vivo blood analysis from the same subjects demonstrated that after the beer consumption period, *E. coli*-stimulated macrophages showed elevated levels of oxidative burst, but no changes in phagocytic activity [89], as well as an improved capacity of PBMCs for the production of IL-2, IL-4, IL-6, IL-10, TNF-α and IFN-γ in both sexes [8]. In this study, the IFN-γ/IL-10 ratio decreased in both women and men, considering that a high IFN-γ/IL-10 ratio has been associated with depressive disorders, moderate beer consumption could have anti-depressive effects without affecting the proinflammatory cytokine production (no changes on TNF- α and IL-6). In addition to this, it has been shown that moderate non-alcoholic beer consumption prevented the acute inflammation caused by aerobic endurance exercise, in this case, alcoholic and non-alcoholic beer prevented the increase in serum IL-6 and the number of circulating leukocytes given after finishing a marathon [91]. 

In the case of subjects with cardiovascular risk, moderate beer consumption increased the level of macrophage-produced microRNAs that have been related to a decreased expression of inflammatory cytokines [92]; accordingly, moderate beer consumption also decreased the blood levels of inflammatory markers IL-5, IL-15, the receptor of IL-6, and the chemokine regulated upon activation, normal T cell expressed and secreted (RANTES) in cardiovascular risk individuals [93]—however, no changes in IL-6 or TNF-α were reported [9].

Studies focused on moderate beer consumption and the effect of the natural compounds found in beers have shown important roles in the modulation of immune responses, beyond anti-inflammatory effects, beer components seem to play a more complex part in the different immune populations, also depending on the type of immune challenge that is presented. Further research is needed to clarify these complex behaviors and incidentally reinforce the benefits of moderate beer consumption.

## 5. Hop Derivatives and Neuroprotection

The prevalence of age-related neurodegenerative diseases is exponentially increased due to a longer life expectancy; currently, there are no effective therapies available to ameliorate their progressive disabling characteristics. The processes that contribute to neurodegeneration include excitotoxicity, the dysfunction of cellular organelles such as mitochondria, and lysosomes favoring an oxidative environment and losing the redox homeostasis which together with an immune imbalance lead to inflammatory signaling and altered responses of glia in the brain. These events converge and contribute to misfolded proteins, protein aggregation, cellular communication damage, and altered—which subsequently induce brain cellular loss. These cellular and molecular processes can be reflected in the cognitive impairment and motor deterioration characteristic of aging and neurodegenerative disorders [96,97,98,99]. Recently, it has been reported that the moderate drinking of beer shows some positive effects by improving cognitive impairments related to several of these neurological and pathological conditions. These effects have been related to the antioxidative and anti-inflammatory properties of the chemical compounds that can be found in the hops, described above. Specifically, these components have shown effects in neurotransmission, redox modulation, and neurogenesis, providing neuroprotection in different experimental models (Figure 3).

One of the earliest indications showing that hop components could act on the central nervous system was the fact that hop cones have been widely used in folk medicine as a tranquilizer, sedative, and anxiolytic, presumably by an effect on γ-aminobutyric acid (GABA)_A_ receptors. This idea was supported by evidence showing that extracts from *H. lupulus* decreased the locomotor activity, increased sleeping time, and overall potentiate the sedative effects induced by pentobarbital or ketamine [100,101,102] through the inhibitory action of GABAergic neurotransmission. In this line, the effect of beer components on GABA_A_ receptors has been studied in *Xenopus oocytes* injected with cRNAs of α_1_ and β_1_ subunits of bovine GABA_A_ receptors to induce their expression. The beer components myrcenol, linalool, geraniol and, 1-octen-3-ol induced a stronger potentiation (2- to 4-fold taking as reference the response elicited by GABA) on the GABA_A_ receptor activation; while hop oils showed a small potentiation (20–50% compared with the GABA response). In the absence of a preliminary GABA stimulus, these compounds did not induce a response on the GABA_A_ receptors, therefore, they did not act as agonists but as enhancers. Among all the compounds examined, myrcenol showed the greatest potentiation on the GABA_A_ receptor using this *Xenopus oocytes* expression system; for this reason, the effect of myrcenol was also tested in vivo where, accordingly, it potentiated the GABA_A_ response extending the sleeping time induced by pentobarbital [103]. On the other hand, contrasting effects were attributed to the hop β-acids since they reduced the GABA-evoked currents in a concentration-dependent manner in cerebellar granule cells [104]. 

Recently, it was described that xanthohumol, isoxanthohumol and 8-prenylnaringenin can displace 3-ethynylbicycloorthobenzoate, a noncompetitive blocker of GABA_A_ receptor, and bind to GABA_A_ receptors with an IC_50_ of 29.7 ± 0.8 µM, 11.6 ±0.7 µM and 7.3 ± 0.4 µM, respectively. This effect showed a positive allosteric modulation of the GABA-induced responses where the highest potency was observed for 8-prenylnaringenin with a relatively higher subtype selectivity for those GABA_A_ receptors containing δ-subunits [105]. The effect of 8-prenylnaringenin suggests that it can directly activate GABA_A_ receptors considering that this hop derivative can displace almost 50% of radioligand binding ([^3^H]Ro 15-4513, an inverse agonist of GABA_A_) [105]. Furthermore, xanthohumol was able to decrease glutamate release from rat hippocampal synaptosomes and reduce miniature excitatory postsynaptic currents in hippocampal slices. This effect was attributed to a presynaptic mechanism, which involved the modulation of GABA_A_ receptors [106].

Considering that xanthohumol inhibits glutamate release, Wang and coworkers studied the effect of this chalcone in a rat model of excitotoxicity induced by an intraperitoneal injection of kainic acid (KA). In this case, xanthohumol administration (10 and 50 mg/kg) effectively reduced glutamate levels, seizure score and neuronal death after KA administration [107]. Furthermore, the mitochondrial alterations (swelling, disruption, and decreased size) induced by KA and observed in the hippocampal CA3 region were prevented by xanthohumol pretreatment; in fact, this compound restored the downregulated mitofusin-2 and consequently prevented the increase in the apoptotic protease activating factor 1 (Apaf-1) and the levels of cleaved caspase-3. In sum, xanthohumol prevents the excitotoxic damage induced by kainic acid inhibiting glutamate release, preserving the mitochondrial functionality, and hence preventing the mitochondrial-dependent apoptotic process [107].

Other studies have investigated the effect of xanthohumol on cognitive performance, senescence, and neuronal differentiation models. In this context, this chalcone has been shown to reduce brain proinflammatory, TNF-α and IL-1β, and proapoptotic (BAD, BAX and AIF) markers in senescence-accelerated prone mice (SAMP8) and to increase the protein expression of neural trophic factor (BDNF), synapsin and synaptophysin [108]. This evidence suggests a positive effect of xanthohumol on cognitive performance in aging or other neurodegenerative diseases. Particularly, it has been observed that a xanthohumol diet (30 mg/kg body weight/day for 8 weeks) improves cognitive flexibility in young mice [109]. Moreover, its derivative isoxanthohumol and the flavanones 8-prenylnaringenin and 6-prenylnaringenin promote neuronal differentiation in fetal progenitor cells [110,111]. Future studies that integrate new tools related to neurogenesis are the next step to decipher the mechanistic role of xanthohumol and isoxanthohumol in this issue.

There are other mechanisms studied through which xanthohumol exerts its neuroprotective effects (Figure 3). The systemic administration of xanthohumol (0.2 and 0.4 mg/kg) reduced (40 and 60%, respectively) the infarct area in cerebral ischemic rats, improving their neurobehavioral deficits. This protective xanthohumol effect was associated with a decrease in inflammatory and apoptotic proteins’ expression such as TNF-α, caspase-3, hypoxia-inducible factor (HIF)-1α, and inducible nitric oxide synthase protein (iNOS) [37]. 

Xanthohumol activates the Nrf2-ARE signaling pathway in PC12 cells, leading to the upregulation of phase II enzymes expression, therefore preventing the neurotoxicity induced by hydrogen peroxide and 6-hydroxydopamine. The Nrf2-signaling pathway activation as a mechanism by which xanthohumol exerts neuroprotection was confirmed in PC12 cells transfected with shRNAs silencing the rat Nrf2 gene (shNrf2s)—as a result, the cytoprotective effect of this chalcone was lost [41]. Additionally, xanthohumol showed anti-inflammatory activity in a murine microglial cell line (BV2 cells) exposed to LPS. Briefly, the pretreatment with xanthohumol (5 µg/mL) decreased the overproduction of NO, as well as the increased expression of iNOS, COX-2, TNFα, IL-1β, and NF-κB, in LPS-stimulated BV2 cells. The inhibitory effect of this chalcone was related to the fact that xanthohumol pretreatment increased the Nrf2 activation and, in consequence, the NQO1 and HO-1 mRNA and the protein levels as well as the GSH levels increased [44]. Moreover, xanthohumol and quercetin prevented the neuronal morphological alterations, thus contributing to the preservation of cell viability while preventing astrogliosis, attributable to chronic exposure to corticosterone. However, the mechanism by which these polyphenols act is different since xanthohumol involves the Nrf2 pathway while quercetin attenuates the activation of the glucocorticoid receptor (GR) [44]. 

Additionally, xanthohumol has shown positive effects on neurodegeneration models, for instance, it reduced Aβ accumulation, APP processing and attenuated tau hyperphosphorylation in a cellular model of Alzheimer’s disease (AD) (murine neuroblastoma N2a cells expressing the human Swedish mutant amyloid precursor protein). Proteomic analysis comparing the proteins in lysates of N2a/APP cells in the presence or absence of xanthohumol showed 21 differentially expressed proteins involved in endoplasmic reticulum (ER) stress, oxidative stress, proteasome molecular systems, and the neuronal cytoskeleton [112]. 

In a different mouse model of AD, in a transgenic 5xFAD mouse, it was found that iso-α-acids ameliorated the pro-inflammatory cytokine production, increased microglial phagocytosis of the β-amyloid peptides, prevented tau phosphorylation and tauopathy, and induced the improvement in both memory and cognitive function [61,62,63]. Similar results have been reported in aged mice fed with iso-α-acids which prevented the age-related cognitive decline by decreasing the levels of pro-inflammatory cytokines and promoted the profile of microglia with an anti-inflammatory phenotype [85]. Moreover, iso-α-acids have shown neuroprotective effects in animal models of brain hematoma through the inhibition of NF-κB signaling, the reduced production of IL-1β and TNF-α, and the promotion of anti-inflammatory microglial activation, which contributed to the improvement in the cognitive functions of these mice [84]. 

On the other hand, hop bitter acids intake improved spatial and object recognition memory functions and increased hippocampal dopamine levels through vagus nerve activation. The memory improvement induced by hop bitter acids is attenuated when vagotomy is performed, suggesting that iso-α-acids activate the vagus nerve and possibly lead to increased hippocampal dopamine levels and consequently they improve memory function [113]. Moreover, the hop bitter acids diet increased the dopaminergic activity associated with stress resilience in a repeated social defeat stress mouse model [114].

The neuroprotective effect of hop components has also been demonstrated in human studies. In a randomized double-blind placebo-controlled study in healthy adults, it was observed that the consumption of matured hop bitter acids (35 mg/day for 12 weeks) improved mental fatigue, mood state, verbal memory retrieval, and anxiety when compared with the placebo group (45–64 years) [115]. Indeed, hop bitter acids improved cognitive function, attention, and mood state in older people (45–69 years) with subjective cognitive decline [116]. 

## 6. Conclusions

Throughout this review, it has been described that beer is one of the most consumed beverages worldwide and its components possess relevant antioxidant and immunomodulatory properties. In particular, hop compounds have demonstrated their effectiveness in improving neurodegenerative processes since they can modulate the cellular redox environment, favoring an antioxidant and anti-inflammatory response, which in turn attenuates the damage of harmful agents or simply prevents physiological alterations due to aging. Based on this evidence, beer consumption and hop compounds represent an excellent research target to prevent or ameliorate cellular processes that lead to neurodegeneration.

## Figures and Tables

**Figure 1 nutrients-13-02000-f001:**
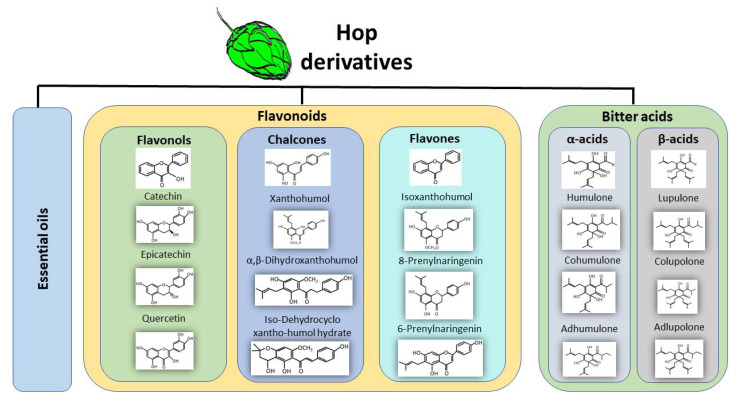
The main derived compounds from hop cones. Hop cones are the main source of bioactive compounds during the brewing process. Hops contribute essential oils, phenolic compounds such as flavonoids which can be further classified into flavonols, chalcones, and flavones. Hop cones contain α-acids and β-acids also known as bitter acids.

**Figure 2 nutrients-13-02000-f002:**
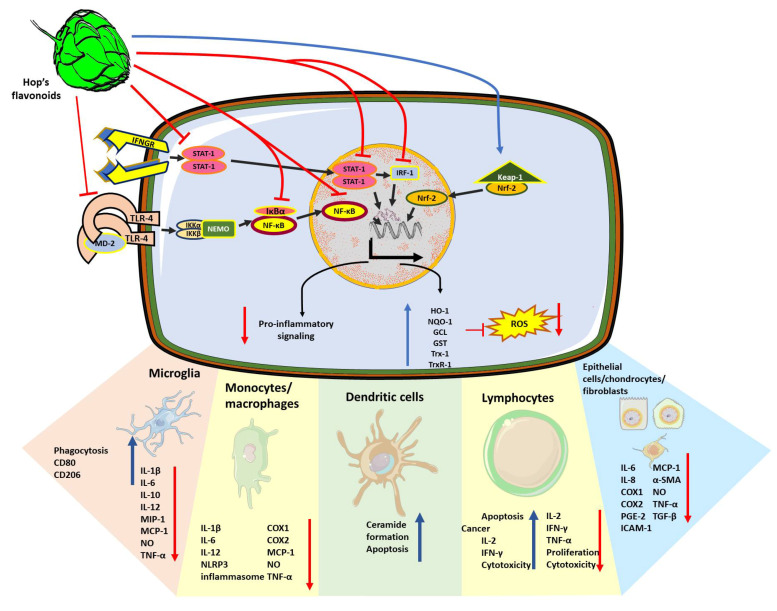
Cellular mechanisms of the anti-inflammatory, immunomodulatory and antioxidant effects exerted by hop compounds. Polyphenols extracted from *H. lupulus* such as xanthohumol, isoxanthohumol and bitter acids can interfere with intracellular immune signaling pathways at different levels. Due to the inhibition of TLR4 activation through preventing the association of TLR4 dimers with the co-stimulatory molecule MD2, NF-κB nuclear translocation was prevented. Hop compounds inhibit NF-κB signaling pathway due to both, the inhibition of TLR4 activation through preventing the association of TLR4 dimers with the co-stimulatory molecule MD2, and in a TLR-independent fashion. Hop compounds can inhibit other pro-inflammatory signaling pathways driven by STAT-1 and IRF-1. Thus, reducing the expression of the subset of pro-inflammatory mediators such as IL-1β, IL-6, IL-8, IL-10, IL-12, TNF-α, inflammasome subunits, nitric oxide, cyclooxygenases, and prostaglandin production, adhesion molecules such as ICAM-1, MCP-1, and MIP-1. Simultaneously, hop compounds promote microglial phagocytic activity together with the switching to M2 phenotype. In different cell types, hop compounds also prevent fibrotic processes by reducing the expression of α-SMA or TGF-β. However, hop compounds also promote the pro-inflammatory and cytotoxic responses from lymphocytes to elicit anti-tumor responses. Finally, in many cases, the anti-inflammatory effects of hop compounds are accompanied by the activation of the antioxidant response regulator, Nrf-2, and the increase in the production of HO-1, NQO-1, GST, GCL, Trx-1 and TrxR-1.

**Figure 3 nutrients-13-02000-f003:**
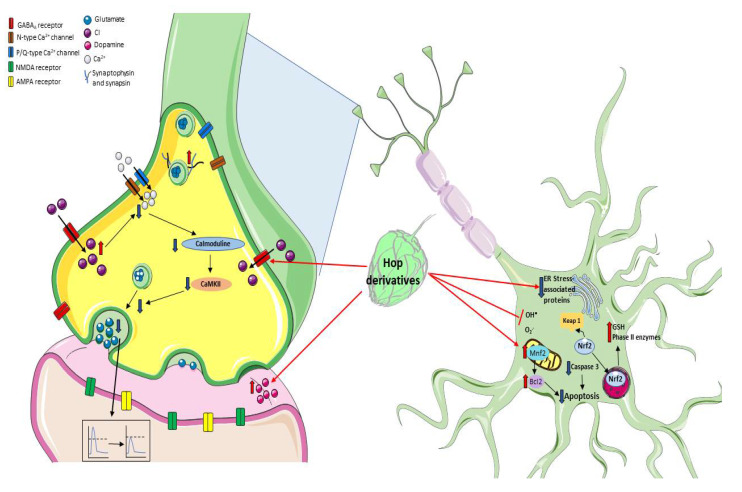
Mechanisms of neuroprotection by prenylflavonoids. It has been described that prenyl flavonoids such as xanthohumol can modulate the GABA_A_ receptors, thus increasing intracellular Cl^−^ concentrations and reducing Ca^2+^ influx, leading to a decrease in glutamate release, and preventing an exacerbated excitotoxic neuronal damage. Additionally, hop metabolites can modulate the redox environment through Nrf2 signaling, regulating the expression of mitochondrial proteins, or preventing oxidative damage by directly scavenging ROS.

**Table 1 nutrients-13-02000-t001:** Effects of beer compounds on immunomodulation.

Beer Compound	In Vitro/In Vivo/Clinical Study	Effect	References
**Xanthohumol or** **Isoxanthohumol**	In vitro		
	Monocytes/macrophages (0.5–20 µM).	- Decreased expression of TLR-4. - Interference with TLR-4/MD-2 association. - Inhibition of NF-κB signaling. - Decreased expression of inflammasome subunits. - Decreased production of IL-1β, IL-6, IL-12, MCP-1, TNF-α, and NO.	[44,66,67,68,69,70]
	Dendritic cells (2–50 µM).	- Increased formation of ceramide. - Activation of caspase-8-mediated apoptosis.	[71]
	T lymphocytes (1.25–40 µM).	- Antiproliferative. - Increased apoptosis. - Reduced lymphocyte cytotoxic activity. - Inhibition of JAK/STAT and NF-ΚB signaling. - Decreased production of IL-2, TNF-α and IFN-γ.	[72,73]
	Mice primary chondrocytes. (10–50 µM).	- Inhibition of NF-κB signaling. - Reduced production of TNF-α, IL-8, PGE-2 and NO.	[74]
	IEC-6 intestinal epithelium (25 µM).	- Inhibition of NF-κB signaling.	[75]
	Primary hepatocytes and hepatic stellate cells (5–10 µM).	- Inhibition of NF-κB signaling. - Decreased production of IL-8 and MCP-1.	[75,76]
	HUVEC cells (0.001–10 µM).	- -Inhibited capillary-like structure formation.	[77]
	In vivo		
	Mice, dextran sodium sulfate-induced colitis (0.1–10 mg/kg orally).	- Inhibition of NF-κB signaling. - Decreased levels of IL-1β, TNF-α and COX2.	[75]
	Liver inflammation, non-alcoholic steatohepatitis, CCl_4_-induced liver injury, ischemia/reperfusion-induced liver injury (0.5% *w*/*w* food).	- Inhibition of NF-κB signaling. - Decreased levels of IL-1α, IL-6, MCP-1, TNF-α and ICAM-1. - Decreased levels of TGF-β, α-SMA and collagen.	[76,78,79]
	Oxalazone-induced inflammation (0.1–5% topically).	- Reduction in ear thickness.	[69]
	LPS-induced lung injury (10–50 mg/kg intraperitoneally).	- Reduced neutrophil count and MPO activity. - Decreased expression of inflammasome subunits. - Reduced levels of IL-1β, IL-6, TNF-α and NO.	[67]
	Skin wound healing (10 mg/L beverage or 50 µM topically).	- Decreased levels of IL-1β, NO and VEGF. - Decreased angiogenesis.	[77,80]
	High-fat diet-induced inflammation (0.01%).	- Reduced levels of circulating IL-1β and TNF-α.	[81]
	Mice, breast cancer (25–50 mg/kg gavage).	- Increased levels of IL-2 and IFN-γ. - Lymphocyte polarization towards TH1 phenotype. - Increased anti-tumor lymphocyte activity. - Reduced tumor volume.	[82]
**8-prenylnaringenin**	In vitro		
	RAW 264.7 macrophages (1–30 µM).	- Inhibition of NF-κB activation. - Decreased expression of TNF-α, iNOS, COX1 and COX2.	[83]
	HUVEC cells (0.001–10 µM).	- Decreased expression of COX2. - Decreased PGI-2 production. - Promoted capillary-like structure formation.	[77,83]
	Spleenic adherent cells (5 µg/mL).	- Decreased production of IL-12.	[69]
	In vivo		
	Rat skin wound healing (50 µM topically)	- Increased level of IL-1β. - Increased angiogenesis.	[77]
**Hop iso-α-acids**	In vitro		
	Primary hepatocytes and hepatic stellate cells (10–20 µg/mL).	- Decreased production of IL-8, ICAM-1, TGF-β and α-SMA. - Increased proliferation.	[64]
	BV-2 microglial cells (1–100 µM).	- Decreased production of NO on LPS-stimulated cells.	[60]
	Mice primary microglia culture. (0.1–40 µM).	- Increased amyloid-β phagocytosis. - Decreased production of TNF-α IL-1β, IL-6, IL-10, IL-12, MIP-1 and MCP-1.	[61,63]
	In vivo		
	Western diet-induced non-alcoholic liver disease mice. (0.5% *w*/*w* in food)	- Decreased levels of IL-1α and TNF-α. - Reduced expression of adhesion molecules. - Decreased levels of TGF-β, MMP-1 and α-SMA. - Decreased lymphocyte infiltration into the liver.	[64]
	Rat intracerebral hemorrhage. (10 mg/kg intraperitoneally).	- Microglial polarization towards M2 phenotype. - Decreased NF-κB expression. - Reduced levels of IL-1β and TNF-α.	[84]
	5xFAD mice (Alzheimer’s experimental model) (0.4–20 mg/kg orally).	- Microglial polarization towards M2 phenotype. - Increased phagocytic activity. - Decreased soluble amyloid-β. - Prevention of amyloid-β deposition. - Decreased production of IL-1β, IL-12 and MIP-1α. - Amelioration of cognitive impairment.	[61,63]
	rTg4510 mice (tauopathy experimental model) (0.5% *w*/*w* in food).	- Reduced levels of IL-1β, IL-12, TNF-α and MIP-1. - Decreased levels of phosphorylated tau.	[62]
	Aged mice. (0.5% *w*/*w* in food).	- Microglial polarization towards M2 phenotype. - Reduced levels of TNF-α and IL-1β. - Reduced level of amyloid-β and glutamate. - Increased level of dopamine. - Improved age-related cognitive impairment.	[85]
**Hop β-acids**	In vivo		
	TPA-induced skin inflammation in mice (5–50 µg/mL topically).	- Inhibition of NF-ΚB signaling. - Decreased pro-inflammatory markers iNOS, COX1 and COX2. - Decreased infiltrated lymphocytes in the skin. - Prevention of tumor formation.	[65]
**Hop bitter acids mix**	In vitro		
	Hepatic stellate cells (10 µg/mL).	- Decreased activation of NF-κB signaling. - Reduced production of MCP-1 and RANTES. - Decreased α-SMA expression.	[58]
	L929sA fibroblasts (0–200 µM).	- Reduced NF-κB activation. - Reduced IL-6 production.	[59]
	In vivo		
	Vagotomized and LPS-intoxicated mice (1–50 mg/kg).	- Reduced blood levels of IL-1β. - Prevention of dendritic spine loss. - Improved depression-like behavior.	[57]
**Hop extract**	In vitro		
	RAW 264.7 macrophages (0.1–100 µg/mL).	- Decreased production of IL-1β, IL-6, TNF-α, MCP-1 and NO.	[52,53]
	THP-1 myeloid cells (0.1–2%).	- Decreased production of IL-10 and TNF-α. - Inhibition of NF-κB activation.	[54]
	PBMCs (3.6–30 µg/mL).	- Decreased COX-2 activity and PGE-2 production.	[55]
	Human nasal epithelial cells (0.1–50 µg/mL).	- Decreased TSLP and TNF-α production.	[56]
**Beer**	In vivo		
	42 mL beer/kg body weight.	- Decreased expression of ICAM, VCAM, NF-κB. - Prevention of atherosclerotic plaque formation.	[50]
	Drinkable beer ad libitum.	- Increased number of anti-tumor reactive lymphocytes. - No difference in tumor growth.	[51]
	Clinical study on healthy subjects Short term (30 min–4 h after single ingestion).		
	Alcoholic beer (355–700 mL).	- Prevented radiation-induced lymphocyte DNA damage (ex vivo). - Reduced lymphocyte cytotoxic activity.	[86,87]
	Long term. (21–45 days of beer consumption).		
	Alcoholic beer (330 mL/day women, 660 mL/day men).	- No changes in cell adhesion molecules. - Increased number of circulating lymphocytes (women). - Increased serum levels of IgM, IgA and IgG. - Increased monocyte oxidative burst capacity (ex vivo). - Increased cytokine production capacity (ex vivo).	[8,88,89]
	Non-alcoholic beer (500–1500 mL daily).	- Unchanged levels of complement system molecules. - Decreased the acute rise of IL-6 and leukocyte after endurance aerobic exercise.	[90,91]
	Clinical study on cardiovascular risk subjects. Long term. (14–28 days of beer consumption).		
	Alcoholic beer (500–660 mL/day men).	- Increased level of macrophage microRNA: miR-145a-5p. - Decreased levels of lymphocyte adhesion molecules and IL-5. - Increased levels of IL-1 receptor antagonist.	[92,93]
	Non-alcoholic beer (500–990 mL/day men).	- Decreased level of macrophage microRNAs: miR-320a-3p, miR-92a-5p, miR-20a-5p and miR-17- 5p. - Decreased level of monocyte and lymphocyte adhesion molecules. - Decreased level of IL-6r, IL-15 and RANTES.	[92,93]

## Data Availability

The data presented in this study are available on request from the corresponding author.

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
