# Peer review of "Redox and Anti-Inflammatory Properties from Hop Components in Beer-Related to Neuroprotection"

_nutrients, 2021, doi:10.3390/nu13062000_

Round 1

Reviewer 1 Report

Overviewing the amount of cell biological and signalling cascade data presented, the title seems to be not appropriate. Neuroprotection comprises only about three pages. What is missing in general is the connection of redox biochemistry and molecular biology/signalling cascade. From a review in this journal I would expect some more detailed explanations compared to pathology oriented journals, where the readers are more familiar with signallisation biochemistry. Then: Why is neuroprotection a antiiflammatory process? That means explaining first the events leading to neurodestruction or at least the damaging processes. A large part of the text is immunology oriented. There are  many neuroimmunological diseases. No literature about the effect of hop compounds on these? In the neuro section at the end, neurodifferentiation processes are mentioned. How do they contribute to the overall improvement of neurodegenerative diseases? We know that estradiol or other neurosteroids have some meaning here. Some minor points: The sentence... colloidal strength of beer due to their protein content.....is misleading. In the literature part: Many authors are quenched by using...et al. This practise cannot be accepted, all of them contributed to the works cited and thus need to be mentioned. Fig. 2&3: Black background is not readerr freindly since there are graphic parts in the black area.

Reviewer 2 Report

The manuscript entitled “Redox and anti-inflammatory properties from hop components in beer-related to neuroprotection” presents an interesting review about the properties of a highly consumed beverage worldwide

The aim of study was to evaluate the effect of hop components on anti-inflammatory properties beer.

The review is a very interesting work and gives a very adequate summary of the properties of the different hop compounds. The manuscript is very well structured and uses an adequate and extensive bibliography.

There are only small errors or doubts that arise at the beginning (pag2)

Second paragraph. The authors name certain risk factors, they could specify some of them.

Third paragraph, line 2. the unit mg/dL is correct? check it.

Third paragraph, line 10 there is a typographical error "incrreased".
